# Nutritional Quality of Gluten-Free Bakery Products Labeled Ketogenic and/or Low-Carb Sold in the Global Market

**DOI:** 10.3390/foods11244095

**Published:** 2022-12-18

**Authors:** Nicola Gasparre, Antonella Pasqualone, Marina Mefleh, Fatma Boukid

**Affiliations:** 1Department of Food and Human Nutritional Sciences, University of Manitoba, Winnipeg, MB R3T 2N2, Canada; 2Department of Soil, Plant and Food Science (DISSPA), University of Bari Aldo Moro, 70121 Bari, Italy; 3ClonBio Group LTD, 6 Fitzwilliam Pl, D02 XE61 Dublin, Ireland

**Keywords:** low carb, high protein, high fiber, bread, cake, biscuits, flour mixes

## Abstract

Gluten-free and ketogenic bakery products are gaining momentum. This study aims to develop a better understanding of the nutritional quality of gluten-free bakery products labeled ketogenic and/or low-carb. For this reason, the products available on the global market that were labeled ketogenic and/or low-carb (*n* = 757) were retrieved and compared to standard gluten-free products (*n* = 509). Overall, nutritionally, no significant differences were found among ketogenic and/or low-carb products due the high intra-variability of each type, but they differed from standard products. Compared to standard products, all ketogenic and/or low carb, irrespective of categories, showed lower carbohydrates that derived chiefly from fibers and, to a lesser extent, from sugars. They also had higher protein contents (*p* < 0.05) compared to standard products. Fats was higher (*p* < 0.05) in ketogenic and/or low-carb baking mixes, savory biscuits, and sweet biscuits than in their standard counterparts. Saturated fats were higher (*p* < 0.05) in low-carb savory biscuits and breads, as well as in ketogenic sweet biscuits than in the same standard products. Overall, median values of the nutrients align with the definition of the ketogenic diet. Nevertheless, several products did not align with any of the ketogenic definitions. Therefore, consumers need to carefully read the nutritional facts and not rely on mentions such as low-cab and ketogenic to make their decision of purchase/consumption.

## 1. Introduction

Bakery products are staple foods worldwide, made basically from wheat flour, salt, and/or sugar. Gluten is a protein complex that is key for the development of bakery products such as bread and cakes, owing to its viscoelastic features [1,2]. Nevertheless, gluten intake might trigger adverse reactions in individuals genetically predisposed to gluten-related allergies and intolerances, and thus they must follow a lifetime gluten-free diet [3,4]. From a technological viewpoint, producing gluten-free products with an equivalent quality to that of gluten-containing counterparts is challenging due to the pivotal role played by gluten in forming a strong protein network that provides structure and allows for gas retention in bread and bakery products [5,6,7]. Gluten-free bakery products are mainly made using flours and starches that have a low functionality compared to wheat flour and thus other ingredients (e.g., hydrocolloids and crosslinking enzymes) are added to create a network similar to gluten [8,9,10]. The main sources of gluten-free flours/starches are rice, corn, potato, and tapioca. Nutritionally, gluten-free products are generally characterized by a high content of carbohydrates (due to starchy ingredients), low protein content, and high calorie content [11,12]. Additionally, gluten-free products are often associated with a high predicted glycemic index owing to their high glycemic load due their starch-based composition, which might be related to serious metabolic issues such as obesity and diabetes [13,14,15]. In recent years, significant research and development have been undertaken to enhance the technological and nutritional quality of gluten-free bakery products by increasingly using wholegrains, pseudocereals, and pulses to raise their protein and fiber contents and decrease that of easily digested carbohydrates [16,17].

Overall, it is assumed that all low carbohydrate strategies aim to reduce the overall intake of carbohydrates, but there is no clear consensus on the definition of a low-carb diet. The ketogenic diet, a specific low-carb diet, has gained immense popularity during the last decade [18]. The ketogenic diet market was valued at USD 10,221.40 million in 2019 and is projected to reach USD 15,266.36 million by 2027 [19]. There are several versions of the ketogenic diet [20]. Classic ketogenic diets are defined as high in fat (90%) and low in carbohydrates (restricting daily carbs to 4%) and proteins (6%) [20]. A modified Atkins diet does not restrict energy content and allows 65% fat, 5% carbohydrate, and 30% protein [21]. Another ketogenic diet is the very low-energy ketogenic diet which allows 13% carbohydrates, 44% fat, and 43% proteins and provides a total energy intake of <800 kcal/day [22]. The consumption of carbohydrate-rich foods, mostly cereal-based foods, fruits, and vegetables is limited during a ketogenic diet, yet it is not a carbohydrate-free diet [20]. Food companies have therefore started to develop, and commercialize, several food products specific for such a diet. The health benefits of the ketogenic diet are speculated to be in association with glycogen depletion and fat mobilization that might result in reducing blood glucose and improving fat burning [23,24,25]. The ketogenic diet has been used against neurologic conditions including autism, dementia, epilepsy, and nerve regeneration [26,27]. However, the long-term effects of a ketogenic diet are not yet fully understood.

From a market perspective, the gluten-free diet moved from a specialty diet to a mainstream market due to consumers associating a gluten-free diet with a healthy lifestyle [28,29]. In fact, the gluten-free bakery market is projected to reach USD 1819.4 million by the end of 2022 and is expected to expand at a compound annual growth rate of 8.2% by 2030 [30]. The demand for gluten-free ketogenic bakery products is increasing exponentially as a “high-fat, low-carbohydrate, adequate-protein” diet strategy is being adopted for weight loss, and treating/preventing diabetes, and neurological disorders [31,32]. Gluten-free ketogenic bakery products have not been researched extensively. For consumers, a deeper understanding of the nutritional facts of bakery products labeled gluten-free, ketogenic, and/or low carbohydrate could help in making a conscious and suitable decision of purchase/consumption. Therefore, the objective of this study was to evaluate the nutritional facts of commercial gluten-free ketogenic and/or low carb bakery products and compare them to standard gluten-free products to identify similarities/dissimilarities. For this reason, this study relied on a market database, Mintel, to be as exhaustive as possible by enabling a concrete illustration of the nutritional quality of products available on the supermarket shelves.

## 2. Materials and Methods

### 2.1. Data Collection

The market search of commercial gluten-free ketogenic and/or low carb bakery products was carried out by consulting the Mintel Global New Product Database (Mintel GNPD-Mintel Group Ltd., London, UK). The Mintel GNPD tracks packaged food and beverage launches in 86 markets worldwide. Each item has detailed product information, such as manufacturer, brand, price, ingredients, claims, and nutritional facts. The search was conducted on 16 September 2022.

The search considered the sub-category of “Bakery”. The inclusion criteria were the date of product launches (1 January 1996 to 16 September 2022), the region (the global market), the presence of gluten-free claim, and the presence of the nutritional facts (i.e., energy, carbohydrates, sugar, protein, fat, saturated fat acids (SFA), fiber, and sodium). Using these settings, three searches were conducted with specific keywords:Search 1 was conducted with the inclusion of “Keto/ketogenic” and “Low/No/Reduced Carb” to retrieve ketogenic and low carb products (K-LC).Search 2 was conducted with the inclusion of “Keto/ketogenic” and the exclusion of products labeled “Low/No/Reduced Carb” to retrieve ketogenic products (K).Search 3 was conducted with the inclusion “Low/No/Reduced Carb” and the exclusion of products labeled “Keto/ketogenic” to retrieve low carb products (LC).Search 4 was conducted with the exclusion of “Keto/ketogenic” and “Low/No/Reduced Carb” mentions to retrieve standard gluten-free bakery products (STD) to be used in the nutritional comparison. This search considered the products launched in the last six months (16 March–16 September 2022) owing to the high number of standard gluten-free products and to capture the most recent launches.

### 2.2. Data Extraction

For all searches, nutritional facts and nutrition claims were collected. The results of all searches were exported to Microsoft Excel (Microsoft Office, Washington, WA, USA). Nutritional facts related to energy (kcal/100 g), total fat (g/100 g), saturated fatty acids-SFA (g/100 g), carbohydrates (g/100 g), sugars (g/100 g), protein (g/100 g), fiber (g/100 g), and sodium (mg/100 g) were retrieved, as well as nutrition claims.

### 2.3. Statistical Analysis

The statistical analysis was carried out using the Statistical Package for Social Sciences software (IBM SPSS Statistics, Version 25.0, IBM Corp., Chicago, IL, USA). Based on Kolmogorov–Smirnov test, the normality of data distribution was rejected, and therefore data were expressed as median values with inter-quartile ranges 25th–75th percentile. Energy and nutrient contents of products were analyzed using Kruskal–Wallis non-parametric one-way ANOVA for independent samples with multiple pairwise comparisons.

## 3. Results and Discussion

### 3.1. Description Analysis

Figure 1 shows the evolution of the number of K-LC, K, and LC bakery products in the global market. In total, the number of products retrieved was higher in LC, followed by K and K-LC. LC products have a longer history. The first low products (low-carb cookies) were launched in 2003 in the US market. Products showed a small peak in 2004 due to new launches in both US and Canadian markets. This aligns with a market report underlining that the low-carbohydrate movement started in North America in 2003–2004, where low-carb products ranked fifth among the most desirable new foods [33]. From 2004 to 2014, fluctuations in the number of launches were observed, to start increasing in a steady way in 2015. K-LC and K products appeared in 2016 and 2017, respectively. These products start to grow exponentially from 2018. This can be attributed to carb-watcher consumers increasing across the globe considering ketogenic, and/or views of low-carb diets as healthier diets and responsible for rapid weight loss. However, the main concern remains in the quality of the ingredients added, and, especially, the fat sources of low carb/ketogenic products. On the other hand, special caution should be taken by consumers who decide to adopt such diets because a high fat and protein diet could induce renal dysfunction on the long term [34,35]. A notable peak was observed in all gluten-free products in 2020 which can be attributed to Coronavirus 2019 pandemic [36,37]. The main boosters can be the health halo surrounding gluten-free diet, as well as the mounting number of consumers choosing this diet for weight management.

With 334 products, LC were the most abundant products (Table 1) due to their established history in the market, followed by K and K-LC. The examined bakery products were classified into five categories: baking flour mixes; bread products; cakes, pastries, and sweet goods; savory biscuits and crackers; and sweet biscuits and cookies. Baking flour mixes markedly prevailed over the other product categories, irrespective of being K, K-LC, or LC. The need for excluding or limiting the introduction of carbohydrates obviously excludes, or limits, all standard bakery products. Baking mixes have therefore been formulated by food companies to substitute regular grain flours, allowing for the domestic production of baked goods. In their K version, these mixes are basically composed of vegetable oils (sunflower or palm oil) absorbed on a cellulose substrate and powdered and added to protein from various sources [32]. In decreasing order, the second category most often labelled K or LC was represented by sweet biscuits and cookies, while bread products were the second category for K-LC. Categories such as sweet biscuits and cookies are widely appreciated by consumers because are very palatable and ready-to-eat, helping to improve diet adherence [32]. Regarding the geographical distribution of the examined products, our results confirmed that North America is the leader in K, K-LC, and LC gluten-free bakery products. The low-carb and ketogenic diets have been promoted for weight loss due to the numerous low-carbohydrate diet books, the over-sensationalism of these diets in the media and by celebrities, and the promotion of these diets in fitness centers and health clubs [38]. The high rates of obesity in North America explain the strong interest of the US consumers, who associated the low-carb diet with health and wellness [39], and thus moved toward K, K-LC, and LC products. The typical US diet is high in carbohydrates, mostly simple, accounting overall for approximately more than 65% of caloric intake [39,40]. Low-carbohydrate diets such the Atkins diet (a version of the ketogenic diet) has fueled the diet industry in North America for years, despite limited scientific evidence about its health benefits and risks [18]. During the last 20 years, weight loss and weight management in the US market witnessed a steady growth and was valued at USD 78 billion in 2019 [41]. It should be also underlined that a gluten-free diet is sometimes adopted, and perceived as an effective diet for weight loss, by consumers without gluten related intolerances/allergies [3,42]. The gluten-free diet has sometimes been associated with weight loss in non-gluten intolerant/allergic people because of improvements in insulin resistance and lipolysis [43,44].

LC and K products are also popular in Latin America as this geographical area is facing major diet-related health problems linked to overweight and obesity among all ages, accompanied by enormous social costs [45]. Therefore, in Latin America consumers are also shifting to these diets with the objective to manage their weight. The traditional Latin American cuisine is rich in complex carbohydrates, micronutrients, fiber, and phytochemicals [46,47]. However, during the last 40 years, Latin American countries have been experiencing a nutrition transition, moving from under- to overweight due to excessive consumption of refined carbohydrates and added sugars [47].

In the Asia-Pacific, a similar pattern was observed, especially for LC products. Over the last 20 years, Asian countries decreased the average carbohydrate intake due to increased prevalence of diabetes [48,49]. The traditional Asian diet is characterized by a high intake of rice, soy-based foods, and fish [50]. The Japanese Diabetes Society recommended a caloric reduction of 25–35 kcal/kg ideal body weight with carbohydrates constituting 50–60% of total energy consumption [51]. A recent review underlines that available evidence suggests there is a strong physiological rationale supporting the role of carbohydrate restriction for the management of Type 2 diabetes without inducing an increased risk of cardiovascular disease [52].

The Middle East and Africa ranked fourth, with a total of 53 products (as the sum of K, K-LC, and LC). Traditionally, the Mediterranean diet adopted in the Middle East and North Africa was one of the healthiest diets, as it is rich in vegetable proteins, fibers, minerals, and vitamins [53,54]. However, due to urbanization and changes in lifestyle, this geographical area too experienced a relatively recent nutrition transition to a diet rich in added sugars, and often lacking in vegetables, fruits, and whole grains [55,56]. This transition, associated with an increased burden of non-communicable diseases [57,58], is expected to strengthen the market of K, K-C, and LC products in the Middle East and Africa.

The European market had the fewest launches of products labeled K, K-LC, and LC. Indeed, low carb and ketogenic claims are not among the permitted claims in Europe (Regulation (EU) No 1047/2012). The retrieved products were mostly marketed in the UK, which has different legislation than the European union. The UK government’s dietary guidelines recommend no more than 55% carbohydrate intake per day [59]. Mostly LC products were observed, only two K and no K-LC.

Additional nutrition claims may apply to K, K-LC, and LC products. LC products—the “historical” and most numerous category—were those labelled with the highest number of additional claims (507 claims for a total of 334 products, e.g., 1.5 claims per product). On the contrary, not all K products were additionally labelled (230 claims found for 248 products). Regarding the type of claim, K-LC chiefly had products with sugar reduction claims, followed by fiber claims. K products were mainly related to sugar, fiber, protein, and trans-fat reduction claims. LC had claims mostly related to sugar reduction, followed by fiber and protein enrichment. The reduction of sugar aligns with the fact that a large number of products were sweet and agrees with the ongoing trend in bakery products, especially in gluten-free products, to reduce sugar as reported in previous market surveys [11]. The focus was to reduce starchy ingredients by substituting them with fiber, protein, and sweeteners to preserve product structure and organoleptic features. For K and K-LC, few products had claims related to fat reduction, as the ketogenic diet was typically defined as high in fat [60]. For all categories, few products had claims related to calories, saturated fat, and sodium reduction.

### 3.2. Nutritional Quality

#### 3.2.1. Bakery Flour Mixes

The nutritional composition of gluten-free flour mixes labeled ketogenic and/or low carb compared to their standard counterparts is displayed in the Figure 2. In terms of energy value, the median of all the product categories stayed around 400 kcal/100 g, and non-significant differences were reported (*p* > 0.05). K, LC, and K-LC flour mixes offered the highest fat amounts (below 30 g/100 g), while the lowest values belonged to the standard ones (*p* < 0.05). Regarding saturated fatty acids, the same trend was shown where standard products contained significantly (*p* < 0.05) less with respect to the gluten-free baking K and/or K-LC flour mixes. To better understand the nutritional variations, valuable help is provided by Appendix A, which contains all the ingredients present in the product ingredient lists. According to Appendix A, high-fat flours from almond, coconut, soybean, and tiger nut appeared most frequently among the ingredients of the K, K-LC, and LC products due to their high content of protein and fat [61,62], unlike the standard bakery flour mixes that, in addition, included also flours from cassava, corn, potato, and rice. Oils from coconut, sunflower, and palm were the most employed fat for these product categories, with butter appearing only in the ingredient lists of the LC products. No significance (*p* > 0.05) was found among the different sodium contents, the value of which was below 500 mg/100 g. The median values of the carbohydrate contents confirmed what was reported on the product packages, standard bakery flour mixes had significant (*p* < 0.05) higher contents (around 65 g/100 g), as opposed to the K, K-LC, and LC products (below 40 g/100 g). The sugar content followed the same tendency, in which amounts below 5 g/100 g characterized the gluten-free K, K-LC, and LC products, whereas significantly (*p* < 0.05) slightly higher contents were found in the standard products. This could be explained by looking at the Appendix A, which shows that ingredients with a high glucose content, such as cane sugar, coconut palm sugar, brown sugar, glucose, and maltodextrin were specially employed for the production of the standard products; no added sugars were reported on the ingredient list of the K products. On the other hand, sweeteners, such as erythritol, stevia, and xylitol were largely employed in the K, K-LC, and LC products to substitute sugars; allulose was the only sweetener used for the production of standard products. The situation changed when products were compared in terms of fiber content. Standard products were significantly (*p* < 0.05) the lowest, while K products occupied the middle position, and K-LC with LC bakery flour mixes held the highest fiber content (around 20 g/100 g). The number of fiber ingredients present in the ingredient list of the K, K-LC, and LC products was much higher compared to that of the standard products. Xanthan gum, psyllium seed husks, inulin, oat fiber, resistant dextrin, guar gum, apple fiber, and cellulose were the most utilized fibers. The adoption of the aforementioned ingredients not only serves for nutritional improvements, making the gluten-free products healthier, but it is also intended for a structuring function within the food matrix, while adhering to the low net carbohydrate requirements [23,63,64].

From a technological standpoint, the gluten absence and the reduced presence of starchy raw materials mandates the use of hydrocolloids, gums, and fiber to recreate a pseudo gluten network with the aim of increasing the gas retention and creating a well-defined crumb structure [8,65]. All the K, K-LC, and LC gluten-free products were significantly (*p* < 0.05) higher in protein content (around 15 g/100 g) compared to their standard equivalents (below 7 g/100 g). Egg white proteins were the only protein used for the manufacture of the standard products. Typically, the protein source in gluten-free products derived from animal sources [23]. However, for the other product categories, beside proteins from animal sources (egg, whey, and milk), plant-based proteins were used, such as pea and rice (Appendix A). Pea and rice proteins are increasingly used due their gluten-free nature and good functionality. Nutritionally, both have limitations in terms of amino acids compared to egg, but the blended cereal–legumes proteins could offer a balanced amino acid profile [66,67]. This can be attributed to the increased interest towards vegan products, and thus K, K-LC, and LC products are considered to fit within this trend in the bakery sector [68,69]. Decreasing carbohydrates, fat, and protein contents resulted increases in K, K-LC, and LC products. By resorting the use of proteins, food manufacturers principally wanted to meet nutritional enhancement and strengthening of the structure, which allows the obtention of gluten-free products with better properties in terms of texture and mouthfeel [70]. Fats rich in saturated fatty acids are increasingly being avoided due to their undesirable health effects, such as increasing the risk of cardiovascular disease and metabolic syndrome [71,72]. Fats play different roles in bakery products, including the promotion of moistness, mouthfeel, and soft texture [73].

#### 3.2.2. Bread Products

Figure 3 displays the nutritional analysis of the gluten-free ketogenic and/or low carb and standard breads. Standard breads had significantly (*p* < 0.05) the highest energy content (Figure 3). No significance (*p* > 0.05) was found among the different fat contents in all the product categories, but LC breads showed the highest saturated fatty acid amounts, significantly higher than in their standard gluten-free counterparts, maybe because of the presence of coconut oil and full fat milk in their formulations (Appendix A). This observation has negative health implications, especially considering that the overall nutritional quality of the fatty fraction of standard gluten-free products is not particularly high, as highlighted by several studies [74,75,76], so it should not be worsened further. For the specific purpose of improving nutritional quality, the use of extra virgin olive oil in gluten-free bread-making has been proposed [77]. For sodium content, statistical analysis showed no significant difference among categories (*p* > 0.05). The analysis of the carbohydrate contents related to the K, K-LC, and LC products highlighted that their median values were below 30 g/100 g, making them significantly (*p* < 0.05) lower than the standard products (above 45 g/100 g). Concerning the fiber content, K-LC and K products reached the highest values (between 12 and 18 g/100 g), followed by LC (10 g/100 g), with the standard gluten-free breads that contained the significantly (*p* < 0.05) lowest fiber content. Fibers from bamboo, oat, as well as rice bran, cellulose, and carob bran were found in the ingredient lists of LC, K, and K-LC breads (Appendix A) and this underlies the versatility of fiber enrichment. Increasing fiber is a proven strategy to increase gluten-free bread’s nutritional features and improve its sensory properties [78]. Small differences were found in sugar content, attributed to the high intra-variability of each group, especially the standard products. This is due to the general tendency of reducing sugar in gluten-free baked goods for health motives [79,80]. White sugar, cane sugar, brown sugar, glucose, rice syrup, and agave syrup were the sugars employed for the standard gluten-free breads, as opposed to maple syrup and glycerol, which were only utilized in the K and LC formulations, respectively (Appendix A). A protein content around of 5 g/100 g positioned the gluten-free standard breads in the last position, whereas K-LC and K breads reached the highest values (above 12 g/100 g). As shown by the Appendix A, proteins from pumpkin seeds and hemp were incorporated in the recipes for the ketogenic and/or low carb breadmaking in addition to protein from eggs, soybean, and pea that were present also in the standard formulations. These proteins are added to raise the nutritional value and to substitute gluten functionality, improving bread properties, such as crumb structure and volume [73,81].

#### 3.2.3. Cakes, Pastries, and Sweet Goods

The nutritional features of the K, K-LC, and LC gluten-free cakes, pastries, and sweet goods are reported in the Figure 4, as well as their standard homologues.

K cakes offered the lowest energy content (around 300 kcal/100 g) compared to K-LC products with the highest energy density (350 kcal/100 g). Statistical analysis exposed no significant (*p* > 0.05) differences among the fat, saturated fatty acids, and sodium contents of the products considered for this study. Concerning the carbohydrate contents, they were significantly (*p* < 0.05) lower in K-LC and K products than in LC and standard products. These latter showed the highest sugar content (around 30 g/100 g), as opposed to their K, K-LC, and LC equivalents, which stayed below 5 g/100 g. Regarding the fiber content, LC cakes, pastries, and sweet goods significantly (*p* < 0.05) contained the highest quantity (above 10 g/100 g), as opposed to the other products included in the study. Reducing sugar in K and/or LC induced the increase of fiber to mimic the functionality of sugar, while sweeteners were added to preserve the taste [82]. This is a general strategy in reduced sugar/sugar-free bakery owing to rising health concerns over the high consumption of sugar [83,84]. These changes explain, in part, the reduction of calories in LC and/or K products. Therefore, these products might fit the requirement not only of celiacs or people following keto and low carb diets, but also of health-conscious consumers looking for a tasty and low caloric cake. According to the Appendix A, only eggs and spirulina extract were used as protein ingredients in the gluten-free standard product formulations, reaching a median value of 5 g/100 g (Figure 4). Products belonging to the K-LC category showed the highest protein content (above 15 g/100 g), followed by LC and K products, respectively, with egg proteins and casein or milk proteins being the main sources of protein.

#### 3.2.4. Savory Biscuits and Crackers

Figure 5 shows that the energy value provided by K, LC, and K-LC products was significantly (*p* < 0.05) higher than that of the standard products. Considering the energy value of fat (9 kcal/g), the fat content boxplots confirmed the previous outcomes, as they showed a similar pattern to energy content. With a fat amount above 40 g/100 g, K products led the group, while the gluten-free standard savory biscuits and crackers provided the lowest values (slightly above 5 g/100 g). The use of fats, mostly from vegetable sources (Appendix A), allowed the obtainment of the significantly (*p* < 0.05) lowest content of saturated fatty acids (around 1 g/100 g) in the regular gluten-free products, as opposed to LC products, in which the median content was around 5 g/100 g, with the upper quartile touching values up to 40 g/100 g. Concerning the carbohydrate content, K-LC, K, and LC products reached significantly (*p* < 0.05) higher median values, around 15, 25, and 35 g/100 g, respectively; standard gluten-free products continued showing significantly (*p* < 0.05) higher carbohydrate contents (around 75 g/100 g). Shifting to the fiber contents, outcomes highlighted that gluten-free products labeled K, K-LC, and LC provided significantly (*p* < 0.05) higher fiber, with K-LC products leading the group. Xanthan and guar gums, as well as inulin, represented the only fiber sources utilized in the formulations of the gluten-free standard savory biscuits and crackers (Appendix A); this could explain their lowest median values (around 5 g/100 g). Dried eggs, whey protein concentrate, milk protein, and hemp protein (Appendix A) were the protein ingredients that most frequently appeared on the ingredient lists of K-LC, K, and LC products, giving them median values ranging from 15 to 20 g/100 g. On the other hand, gluten-free standard products contained just eggs, with a protein content of 5 g/100 g. No significant (*p* > 0.05) differences were found among the different products regarding the amounts of sodium (around 500 mg/100 g) and sugar (about 3 g/100 g).

#### 3.2.5. Sweet Biscuits and Cookies

Outcomes from the nutritional analysis of the gluten-free sweet biscuits and cookies labeled ketogenic and/or low carb, as well as gluten-free standard products, are summarized in Figure 6.

The energy value box plots revealed a significantly (*p* < 0.05) little higher value offered by the standard products, as compared to those of K-LC, K, and LC. Despite this, standard products (~20 g/100 g) contained significantly (*p* < 0.05) less fat than LC (~20 g/100 g), K (~30 g/100 g), and K-LC (40 g/100 g), with these last two product categories showing the highest content of saturated fatty acids. In all the analyzed products, the median values for the sodium content were below 400 mg/100 g, with K and LC products having the highest amounts. These results were dependent on the only additive present in the ingredient lists, sodium hydrogen carbonate (Appendix A). In the presence of some acids, this commonly used chemical leavening agent reacts with them, releasing carbon dioxide [64] and ensuring proper biscuit porosity. K-LC, K, and LC gluten-free products presented a median value of ~35 g/100 g of carbohydrates and 5 g/100 g of sugar. The total energy coming from sugar is lower than 5%, confirming that K-LC, K, and LC gluten-free sweet biscuits and cakes were suitable for the low carb dietary regimes, unlike gluten-free regular products that significantly (*p* < 0.05) contained the highest amounts of carbohydrates (~65 g/100 g) and sugars (20 g/100 g). A different pattern was presented when fiber content was analyzed. In particular, standard products were the poorest (~5 g/100 g), while the K, K-LC, and LC products significantly (*p* < 0.05) included up to twice the amount of fiber. The resorting to the use of eggs, egg whites isolate, soy protein, whey protein concentrate, milk protein, and whey protein isolate (Appendix A) contributed to increase the protein content in the LC products. Hence, LC products significantly (*p* < 0.05) had the highest content (15 g/100 g), while gluten-free standard sweet biscuits and cookies only reached ~5 g/100 g.

## 4. Conclusions

Consumers’ dietary patterns are changing and the food industry is trying to meet their expectations by launching new products. In the field of gluten-free foods, K, LC, and K-LC bakery products are conquering an important slice of the market. Up to now, no research has been carried out to analyze the global market of gluten-free bakery products labelled as ketogenic and/or low carb. The broad view, offered by this study, pointed out the pivotal role of North America in driving the global market for these food products. Baking flour mixes, bread products, cakes, pastries and sweet goods, savory biscuits and crackers, as well as sweet biscuits were the main categories forming the world market of gluten-free products labelled K, LC, and K-LC.

Overall, nutritionally, no significant differences were found among K, K-LC, and LC products due the high intra-variability of each type, but they differed from the standard products. A common trend was observed in the majority of the product categories analyzed: compared to their standard counterparts, gluten-free K, LC, and K-LC products contained higher levels of fiber and protein, while carbohydrate and sugar contents were lower. The fat content was significantly higher in K, LC, and K-LC baking mixes, savory biscuits, and sweet biscuits than in their regular gluten-free homologous products. Moreover, saturated fatty acids were significantly more abundant in LC savory biscuits and LC breads, as well as in K and K-LC sweet biscuits, compared to gluten free regular products of the same categories. While the higher fiber content is an obviously positive nutritional feature, the higher amount of saturated fatty acids constitutes a potential red flag for human health, especially when consumed for extended periods of time.

These findings suggest that the prolonged consumption of these new product categories always requires prior approval by health specialists. On the other hand, this research will open up a new scenario, which could be valuable in order to intensify the collaboration between researchers and the food industry with the aim of improving the nutritional quality of gluten-free ketogenic and/or low carb bakery products. At the same time, the nutritionally questionable composition observed in some products raises the need for special attention to the content of nutrients whose excessive intake has a negative effect on health, such as saturated fatty acids, in foods that are commonly perceived as healthy by the consumers. Consumers are invited to thoroughly read the ingredients and nutritional facts of these products before purchase.

## Figures and Tables

**Figure 1 foods-11-04095-f001:**
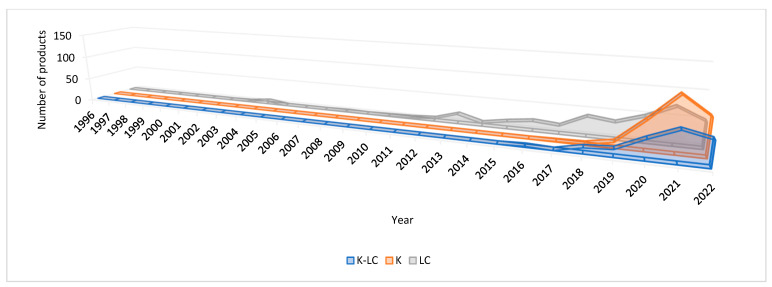
Evolution of the number of gluten-free bakery products labeled ketogenic low carb (K-LC), ketogenic (K), and/or low carb (LC).

**Figure 2 foods-11-04095-f002:**
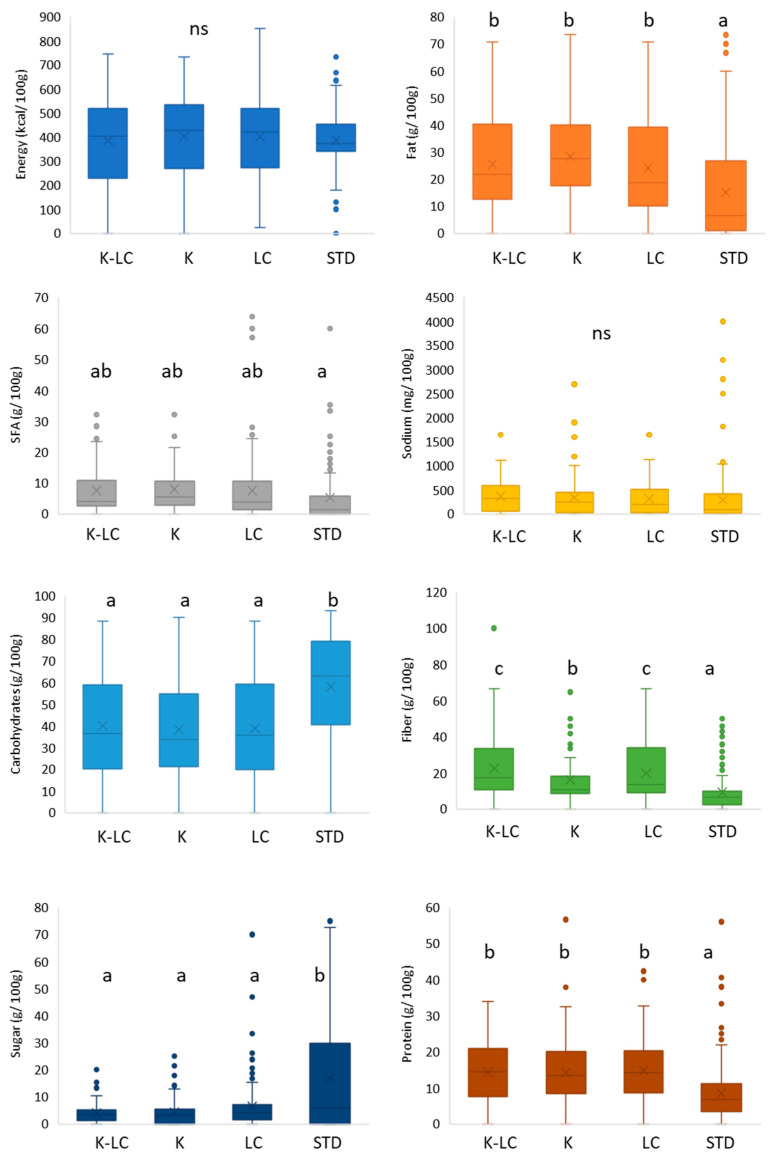
Nutritional composition of gluten-free flour mixes labeled ketogenic and/or low carb. K-LC: ketogenic and low carb; K: ketogenic; LC: low carb; STD: standard; different letters indicate significant difference at *p* < 0.05; ns: non-significant; the box-plot legend: the box is limited by the lower (Q1 = 25th) and upper (Q3 = 75th) quartile; the median is the horizontal line dividing the box; whiskers above and below the box indicate the 10th and 90th percentiles; outliers: are the points outside the quartile 10–90th percentiles.

**Figure 3 foods-11-04095-f003:**
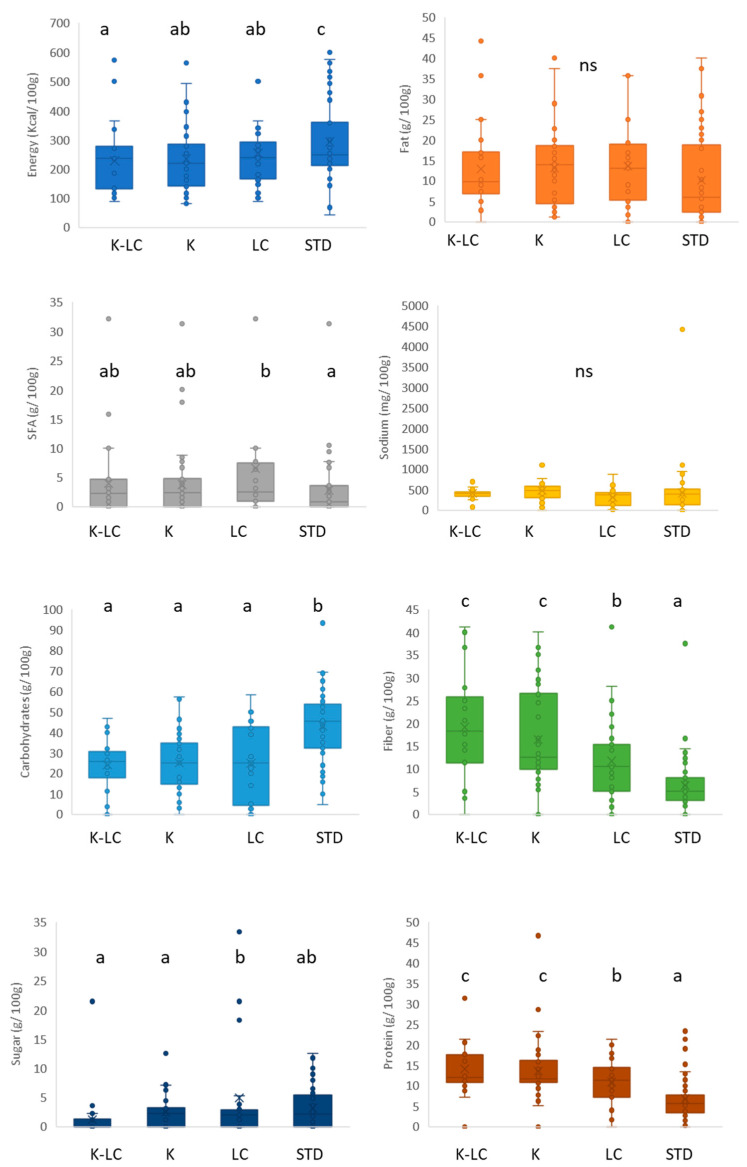
Nutritional composition of gluten-free bread products labeled ketogenic and/or low carb. K-LC: ketogenic and low carb; K: ketogenic; LC: low carb; STD: standard; different letters indicate significant difference at *p* < 0.05; ns: non-significant; the box-plot legend: the box is limited by the lower (Q1 = 25th) and upper (Q3 = 75th) quartile; the median is the horizontal line dividing the box; whiskers above and below the box indicate the 10th and 90th percentiles; outliers: are the points outside the quartile 10–90th percentiles.

**Figure 4 foods-11-04095-f004:**
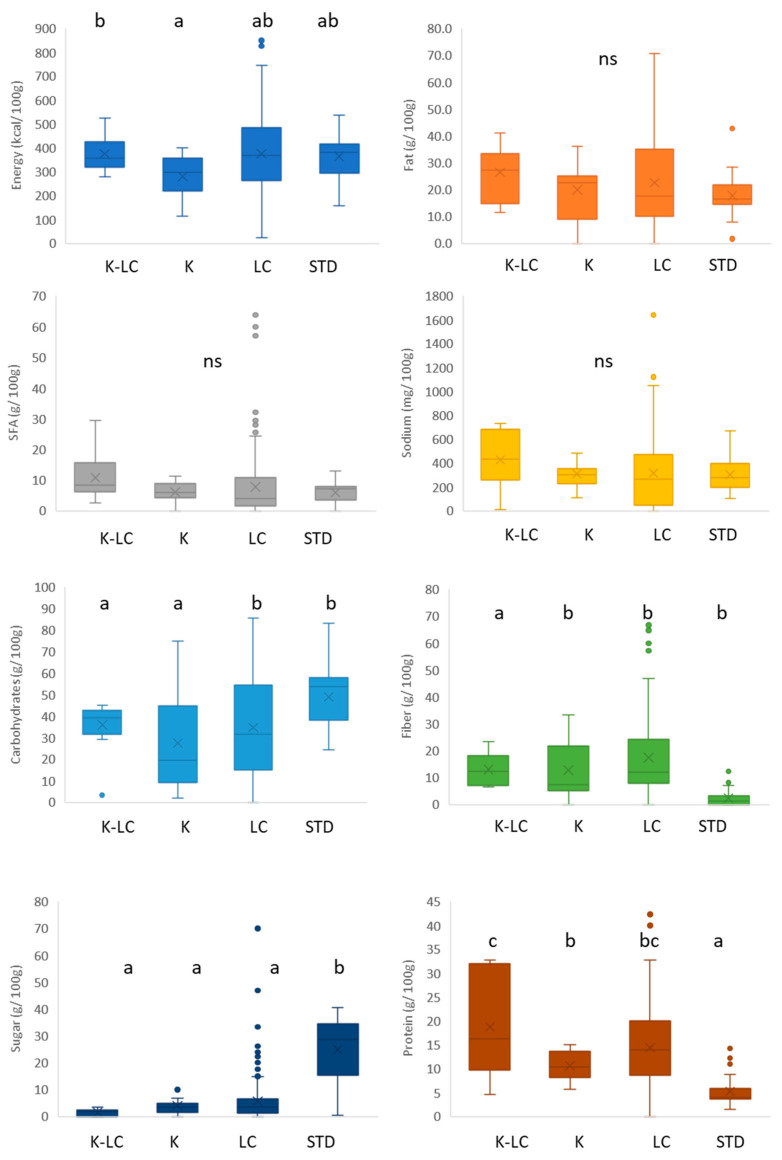
Nutritional composition of gluten-free cakes, pastries and sweet goods labeled ketogenic and/or low carb. K-LC: ketogenic and low carb; K: ketogenic; LC: low carb; STD: standard; different letters indicate significant difference at *p* < 0.05; ns: non-significant; the box-plot legend: the box is limited by the lower (Q1 = 25th) and upper (Q3 = 75th) quartile; the median is the horizontal line dividing the box; whiskers above and below the box indicate the 10th and 90th percentiles; outliers: are the points outside the quartile 10–90th percentiles.

**Figure 5 foods-11-04095-f005:**
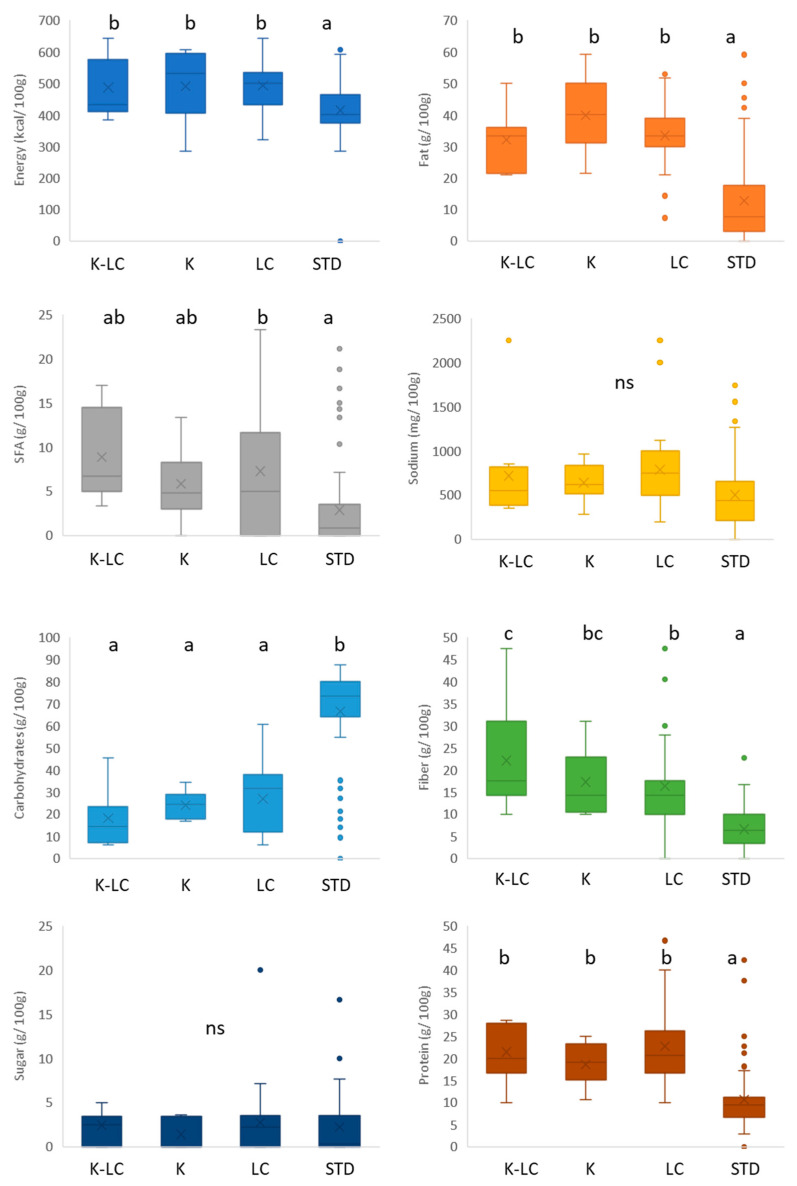
Nutritional composition of gluten-free savory biscuits and crackers labeled ketogenic and/or low carb. K-LC: ketogenic and low carb; K: ketogenic; LC: low carb; STD: standard; different letters indicate significant difference at *p* < 0.05; ns: non-significant; the box-plot legend: the box is limited by the lower (Q1 = 25th) and upper (Q3 = 75th) quartile; the median is the horizontal line dividing the box; whiskers above and below the box indicate the 10th and 90th percentiles; outliers: are the points outside the quartile 10–90th percentiles.

**Figure 6 foods-11-04095-f006:**
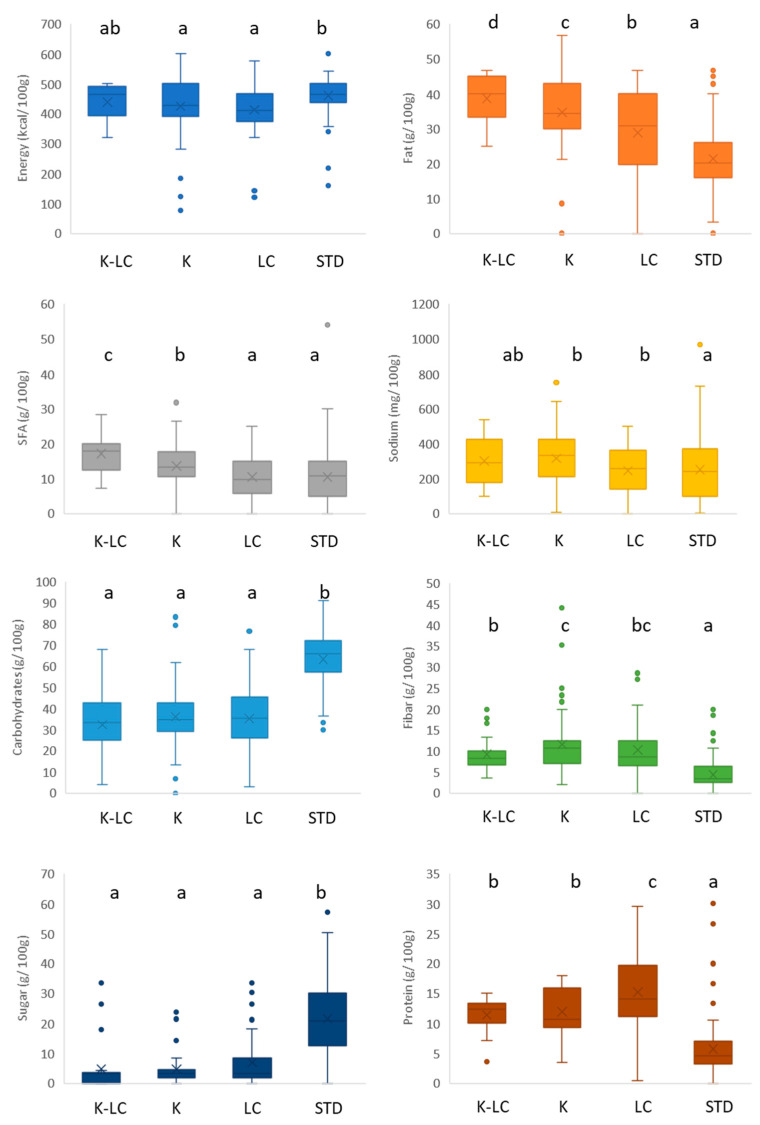
Nutritional composition of gluten-free sweet biscuits and cookies labeled ketogenic and/or low carb. K-LC: ketogenic and low carb; K: ketogenic; LC: low carb; STD: standard; different letters indicate significant difference at *p* < 0.05; ns: non-significant; the box-plot legend: the box is limited by the lower (Q1 = 25th) and upper (Q3 = 75th) quartile; the median is the horizontal line dividing the box; whiskers above and below the box indicate the 10th and 90th percentiles; outliers: are the points outside the quartile 10–90th percentiles.

**Table 1 foods-11-04095-t001:** Categorization of gluten-free bakery products labeled ketogenic (K) and/or low-carb (K-LC, LC).

Criteria	Segmentation	K-LC	K	LC
Type of product	Baking flour mixes	89	93	173
	Bread products	30	53	36
	Cakes, pastries, and sweet goods	15	18	28
	Savory biscuits and crackers	11	10	35
	Sweet biscuits and cookies	27	74	62
	Total	172	248	334
Region	North America	101	186	161
	Latin America	23	42	45
	Asia Pacific	33	13	80
	Middle East and Africa	15	5	33
	Europe	0	2	15
	Total	172	248	334
Nutrition claim *	Low/reduced/no added/free sugar	121	123	219
	High/added fiber	36	41	105
	Low/no/reduced trans fat	11	22	25
	High/added protein	14	12	77
	Low/no/reduced fat	2	11	25
	Low/no/reduced calories	7	11	30
	Low/no/reduced saturated fat	3	4	6
	Low/no/reduced sodium	1	6	20
	Total	195	230	507

* More than one can apply.

## Data Availability

The data presented in this study are available on request from the corresponding author.

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
