# Peer review of "Nutritional Quality of Gluten-Free Bakery Products Labeled Ketogenic and/or Low-Carb Sold in the Global Market"

_foods, 2022, doi:10.3390/foods11244095_

Round 1

Reviewer 1 Report

Article: “Nutritional quality of gluten-free bakery products labeled ketogenic and/ or low-carb sold in the global market”

General remarks

The presented topic is interesting and important for the coeliac as well as the non-coeliac population. The manuscript gives us valuable information about the current offer on the GF product market. Performed analyses are relevant, and the obtained results are properly discussed, hence this article would contribute to the existing literature.

The present research addresses the question of availability and quality of GF bakery products labelled ketogenic and/ or low-carb. The investigated topic is interesting and the present results can give valuable information about the current state on the GF market regarding these specific products addressing this specific gap in the field. Considering that research addressing the nutritional quality of GF products are scarce and their importance was recognised recently, this manuscript will contribute in widening the information about nutritional quality of GF products available on the market especially the this specific kind of it such as ketogenic and/ or low-carb GF products. The conclusions are properly based on the evidences obtained in the presented research. The used references are appropriate.

Specific comments

Maybe the authors should consider comparing the obtained results with the results obtained in studies addressing the nutritional enrichment of GF bakery products by incorporation of dietary fibres, proteins, bioactive compounds.

Page 9, Figure 3.  Please correct ener to energy on the graph.

Author Response

Reviewer 1

General remarks

The presented topic is interesting and important for the coeliac as well as the non-coeliac population. The manuscript gives us valuable information about the current offer on the GF product market. Performed analyses are relevant, and the obtained results are properly discussed, hence this article would contribute to the existing literature.

The present research addresses the question of availability and quality of GF bakery products labelled ketogenic and/ or low-carb. The investigated topic is interesting and the present results can give valuable information about the current state on the GF market regarding these specific products addressing this specific gap in the field. Considering that research addressing the nutritional quality of GF products are scarce and their importance was recognised recently, this manuscript will contribute in widening the information about nutritional quality of GF products available on the market especially the this specific kind of it such as ketogenic and/ or low-carb GF products. The conclusions are properly based on the evidences obtained in the presented research. The used references are appropriate.

The authors thank the reviewer for the positive feedback.

Specific comments

Maybe the authors should consider comparing the obtained results with the results obtained in studies addressing the nutritional enrichment of GF bakery products by incorporation of dietary fibres, proteins, bioactive compounds.

The authors thank the reviewer for the suggestions. However, this suggestion does not align with the scope of the work.

Page 9, Figure 3.  Please correct ener to energy on the graph.1

Figure 3 was corrected.

Reviewer 2 Report

I enjoyed reading the paper although the english should be improved. I think that authors should also include data and references related to the phytochemicals associated to keto and low carbohydrate diets including prebiotics effects of soluble fibers. If this information is included the paper will be more valuable. 

I recommend to edit or modify

Line 11: are gaining momentun.

Line 12: to better understand the n utritional quality ......

Line 17: lower carbohydrates

Line 18: from sugars

Line 37: and other bakery items (5-7)

Line 143: gluten-free diets for .........

Line 200: The Mediterranean diet 

Line 203: richer in added sugars

Line 207. The European market ......

Line 264: fiber and cellulose

Line 301: No significancy was found

Line 309: differences among

Lines 348-350: Need to rephrase the sentence. It is not well constructed and difficult to understand

Line 384: guar gums, .......

Line 390: g/100g. On the other hand, gluten-free.........

LIne 404: are summarized in Figure 6.

Line 635: delete "Vol" 

Author Response

Reviewer 2

I enjoyed reading the paper although the english should be improved. I think that authors should also include data and references related to the phytochemicals associated to keto and low carbohydrate diets including prebiotics effects of soluble fibers. If this information is included the paper will be more valuable.

The authors are grateful for the positive feedback.

I recommend to edit or modify

All the comments were considered in the revised manuscript using track change.

Line 11: are gaining momentun.

Sentence was written in the correct way.

Line 12: to better understand the nutritional quality ......

We have decided to leave the sentence as it is.

Line 17: lower carbohydrates

correction was made in line 17

Line 18: from sugars

We thank the reviewer for the comment. The word has been corrected accordingly.

Line 37: and other bakery items (5-7)

We prefer keeping it without modifications. 

Line 143: gluten-free diets for .........

To make the sentence clearer we have modified it as follows:

“The main boosters can be the health halo surrounding gluten-free diet, as well as the mounting number of consumers choosing this diet for weight management.” (Line 142-143)

Line 200: The Mediterranean diet

We are thankful to the reviewer for the indication, and we have modified the sentence as follows:

“Traditionally, the Mediterranean diet adopted in Middle East and North Africa was one of the healthiest diets as it is rich in vegetable proteins, fibers, minerals and vitamins [53,54].” (Line 200)

Line 203: richer in added sugars

Sentence has been modified accordingly.

Line 207. The European market ......

Sentence has been modified accordingly.

Line 264: fiber and cellulose

Sentence has been modified accordingly.

Line 301: No significancy was found

We thank the reviewer for the indication. The word has been corrected accordingly

Line 309: differences among

The sentence was corrected as follows:

“For sodium content, statistical analysis showed no significant difference among categories (p > 0.05).” (line 308-309)

Lines 348-350: Need to rephrase the sentence. It is not well constructed and difficult to understand

We have reformulated the sentence as follows:

“K cakes offered the lowest energy intake (around 300 kcal/100 g) compared to K-LC products having highest energy density (350 kcal/100 g).” (line 350-352)

Line 384: guar gums, .......

Sentence has been modified accordingly.

Line 390: g/100g. On the other hand, gluten-free.........

Sentence has been modified accordingly

LIne 404: are summarized in Figure 6.

Sentence has been modified accordingly.

Line 635: delete "Vol"

The reference was corrected

Reviewer 3 Report

Introduction: What are the rules and criteria in each country in which these claims are permitted, by serving or 100g/100ml?

Please confirm the statistical letters among treatments in all the figures (ex. SFA from Figure 2, Sugar in Figure 3...)

The similarity of carbohydrate contents between LC and STD gluten-free cakes, pastries, and sweet goods should be more explored since low-carb products should not have similar carbs in comparison to their standard counterparts. 

L.280 Higher or lower? 

Author Response

Reviewer 3

Introduction: What are the rules and criteria in each country in which these claims are permitted, by serving or 100g/100ml?

Low-carb and ketogenic are not an allowed claims by regulatory authorities such as FDA, EFSA and so on. They do not follow any official country regulation.

Please confirm the statistical letters among treatments in all the figures (ex. SFA from Figure 2, Sugar in Figure 3...)

We do confirm that the letters are corrected

The similarity of carbohydrate contents between LC and STD gluten-free cakes, pastries, and sweet goods should be more explored since low-carb products should not have similar carbs in comparison to their standard counterparts.

Totally agree with the reviewer, this is exactly the objective of the work to understand if the nutritional quality of marketed products labeled K and/ or LC aligns with these labels or not in comparison to standards gluten- free products. As the reviewer underlined, the answer is not always, commercial products are so variable and thus not products align with the labels. This point was highlighted throughout the manuscript and the conclusion.

L.280 Higher or lower?

Outliers: are the points outside the quartile 10–90th percentiles on both sides.

Round 2

Reviewer 2 Report

Thanks for considering all my suggestions. The manuscript is much better